# Investigation of DHA-Induced Regulation of Redox Homeostasis in Retinal Pigment Epithelium Cells through the Combination of Metabolic Imaging and Molecular Biology

**DOI:** 10.3390/antiox11061072

**Published:** 2022-05-28

**Authors:** Giada Bianchetti, Maria Elisabetta Clementi, Beatrice Sampaolese, Cassandra Serantoni, Alessio Abeltino, Marco De Spirito, Shlomo Sasson, Giuseppe Maulucci

**Affiliations:** 1Department of Neuroscience, Biophysics Sections, Università Cattolica del Sacro Cuore, Largo Francesco Vito, 1, 00168 Rome, Italy; giada.bianchetti@unicatt.it (G.B.); cassandra.serantoni@unicatt.it (C.S.); alessio.abeltino@unicatt.it (A.A.); marco.despirito@unicatt.it (M.D.S.); 2Fondazione Policlinico Universitario “A. Gemelli” IRCCS, 00168 Rome, Italy; 3Institute of Chemical Sciences and Technologies “Giulio Natta” (SCITEC)—CNR, Largo Francesco Vito, 1, 00168 Rome, Italy; elisabetta.clementi@scitec.cnr.it (M.E.C.); beatrice.sampaolese@scitec.cnr.it (B.S.); 4Faculty of Medicine, Institute for Drug Research, The Hebrew University, Jerusalem 911210, Israel; shlomo.sasson@mail.huji.ac.il

**Keywords:** diabetic retinopathy, oxidative stress, docosahexaenoic acid (DHA), blood-retinal barrier, retinal diseases, therapeutic approach, retinal pigment epithelium, human retinal pigment epithelium cells (ARPE-19)

## Abstract

Diabetes-induced oxidative stress leads to the onset of vascular complications, which are major causes of disability and death in diabetic patients. Among these, diabetic retinopathy (DR) often arises from functional alterations of the blood-retinal barrier (BRB) due to damaging oxidative stress reactions in lipids, proteins, and DNA. This study aimed to investigate the impact of the ω3-polyunsaturated docosahexaenoic acid (DHA) on the regulation of redox homeostasis in the human retinal pigment epithelial (RPE) cell line (ARPE-19) under hyperglycemic-like conditions. The present results show that the treatment with DHA under high-glucose conditions activated erythroid 2-related factor Nrf2, which orchestrates the activation of cellular antioxidant pathways and ultimately inhibits apoptosis. This process was accompanied by a marked increase in the expression of NADH (Nicotinamide Adenine Dinucleotide plus Hydrogen) Quinone Oxidoreductase 1 (Nqo1), which is correlated with a contextual modulation and intracellular re-organization of the NAD+/NADH redox balance. This investigation of the mechanisms underlying the impairment induced by high levels of glucose on redox homeostasis of the BRB and the subsequent recovery provided by DHA provides both a powerful indicator for the detection of RPE cell impairment as well as a potential metabolic therapeutic target for the early intervention in its treatment.

## 1. Introduction

Micro and macro-vascular diseases are major causes of disability and death in patients with diabetes mellitus that have been associated with increased risks for the development of coronary, cerebrovascular, and peripheral arterial disease [1,2]. The pathophysiology of vascular disease in diabetes involves abnormalities in endothelial, vascular smooth muscle cell proliferation and impaired platelet function. The metabolic abnormalities that characterize diabetes, such as hyperglycemia, increased blood free fatty acids content, peripheral insulin resistance [3], decreased bioavailability of nitric oxide (NO), impairments in the lipid turnover [4], increased oxidative stress [5,6], disturbances of intracellular signal transduction, and activation of receptors for advanced glycated end products (AGEs) [7,8,9]. In addition, coagulator functions are impaired due to abnormal platelet function and increased production of several prothrombotic factors. These abnormalities contribute to the cellular events that can alter the normal morphology and functioning of the vasculature that result in progressive organ damage, dysfunction and, ultimately, failure, leading to the development of complications. Microvascular complication, involving small vessels such as capillaries are involved in peripheral complications such as neuropathy, nephropathy and retinopathy [10]. On the other hand, macrovascular complications of large vessels such as arteries and veins, include cardiovascular disease, stroke, and peripheral vascular disease. A better understanding of the mechanisms leading to vascular dysfunction may reveal new strategies to mitigate the morbidity and mortality in patients with diabetes by targeting the factors leading to the onset of diabetes-related vascular complications [11].

Among the several chronic hyperglycemia-induced effects, including inflammation and angiogenesis, increased levels of intracellular reactive oxygen species (ROS) are acknowledged as major detrimental factors [12,13]. ROS are radicals, ions, or molecules characterized by the presence of a single unpaired electron in their outermost shell, a characteristic that makes them highly reactive. Cellular ROS can be endogenously produced as a byproduct of oxidative phosphorylation in mitochondria or may arise from interactions with an exogenous source. Oxidative stress is considered a major contributing factor leading to the etiology of diabetic retinopathy (DR), which is characterized by ischemic microvascular retinal disease and neurodegeneration, being the main cause of blindness worldwide [10,14]. In particular, oxidative stress, being responsible for lipids, proteins, and DNA damage, leads to functional alterations of the blood–retinal barrier (BRB), a tight and restrictive physiological barrier that regulates water, nutrients, ions, and waste products in and out of the retinal compartment [15] and whose disruption has been strictly related to the development of retinal diseases. Retinal pigment epithelium (RPE) plays a pivotal role in the BRB, establishing a major metabolite supply-line for photoreceptors and several support functions, which are essential for cells’ survival and proper function in the visual process [16]. Since RPE cells function both as a selective barrier and an active regulator of the overlying photoreceptor layer, they have become a useful and reliable model for studying diabetes-induced functional alteration of the BRB [17].

Interestingly, among the still existing but less explored factors that contribute to BRB disruption and DR onset, metabolic alterations of the lipid turnover may have an important detrimental effect. According to the results obtained in several clinical trials, a positive association between plasma lipid levels and diabetic retinopathy has been shown [18,19,20], whereas intervention studies with ω3 polyunsaturated fatty acids (PUFAs) appeared to decrease the risk of development and progression of DR [21]. Among the different ω3 PUFAs, docosahexaenoic acid (DHA) has been widely investigated as a potential protective agent, due to its potential anti-inflammatory and antioxidant effects produced in retinal cells [22,23,24,25].

This study aimed to provide functional and molecular characterizations of ARPE-19 cells by combining metabolic imaging and molecular biology investigations. Metabolic imaging allows visualizing the intrinsic fluorescence of the reduced forms of both redox cofactors, Nicotinamide Adenine Dinucleotide (NADH) and Nicotinamide Adenine Dinucleotide Phosphate (NADPH) that can be used to assess the abundance and redox state of these separate pools in living cells [26]. NADH and NADPH are essential enzyme cofactors serving as a major metabolic electron carrier, which participates in the regulation of cellular energetic homeostasis as well as in signaling pathways. Being able to act as an electron donor or acceptor, NAD exists in a reduced (NADH) or oxidized (NAD+) form, used to support energy metabolism in the cytosol and within mitochondria [27,28,29]. Since NADPH, the phosphorylated form of NADH, cannot spectrally be differentiated from NADH, the convention is to describe the mixed intensity signal as NAD(P)H. NAD(P)H exhibits autofluorescence in its reduced form, whereas NAD(P)+ is not fluorescent. This enables microscopic determination of the NAD(P)H redox state by measuring NAD(P)H fluorescence intensity [27,30]. This method is powerful since it is a label-free approach and has a general in-vivo suitability, as demonstrated in numerous tissues such as the brain [31], cochlea [32], and skin [33]. On the other hand, molecular biology techniques allow highlighting pathways associated with redox homeostasis by measuring cellular functions and activation of signaling pathways. Here, we focused on the impairments of redox homeostasis induced by high levels of glucose on RPE cells and the ameliorating effects of DHA before cell viability is compromised, paving the way for a deeper investigation of metabolic approaches for the treatment of DR.

## 2. Materials and Methods

### 2.1. Cells Culture and Treatments

ARPE-19 cells, purchased from the American Type Cell Culture (ATCC–CRL–2302, Manassas, VA, USA), were cultured in Advanced DMEM/F12 basal medium (Thermo Fisher Scientific, Inc., Waltham, MA, USA), containing a physiological concentration of Glucose (5 mM), supplemented with 20% Fetal Calf Serum (FCS, Merck Life Science S.r.l., Milano, Italy) and 100 U/mL penicillin–streptomycin (Gibco™, Thermo Fisher Scientific, Inc., Waltham, MA, USA). Cells were kept in a humidified environment containing 5% CO_2_, until 80% confluence was achieved. Then, the cells were sub-cultured at an appropriate density according to each experimental procedure. To investigate the effect of the high glucose concentrations on ARPE-19 viability and functionality, cultured cells were treated with a 50 mM concentration of D-Glucose 24 h after cell seeding. Twenty hours after sugar administration, ARPE-19 cells were treated with docosahexaenoic acid (DHA) for a further 16 h (Cayman Chemical, Ann Arbor, Michigan, MI, USA—50 mg in 200 µL ethanol). Prior to use, the 0.7 M stock solution of DHA was complexed to Fatty Acid-Free Bovine Serum Albumin (FAF-BSA, Merck Life Science S.r.l., Milano, Italy) at a FA:BSA ratio of 2:1. A total of 2 mg of FAF-BSA were dissolved for each mL of sterile Phosphate Buffered Saline (PBS, Gibco™, Thermo Fisher Scientific, Inc., Waltham, MA, USA) to obtain a 0.003 M stock solution. The FA-BSA conjugated solution was then diluted in complete medium to reach the required physiological concentration (60 µM) of DHA.

All solutions were freshly prepared before each experiment.

### 2.2. Cell Viability

For the determination of cell viability, ARPE-19 cells were seeded into 96-well plates (Greiner Bio-One, Kremsmünster, Austria) at a density of 2.5 × 10^4^ cells/well and treated accordingly, as described above. Cells survival was evaluated by the 3-[(4,5-dimethylthiazol-2-yl)-5,3-carboxymethoxyphenyl]-2-(4-sulfophenyl)-2H tetrazolium inner salt reduction assay (MTS). The MTS assay (Promega Srl-Padova-Italy) [34] provides a sensitive measurement of the normal metabolic status of cells, which reflects early changes in the cellular redox homeostasis. The intracellular soluble formazan produced by the cellular reduction of the MTS was determined by recording the absorbance of each well of the plate using the BioTek Cytation cell imaging multimode microplate reader (BioTek U.S., Winooski, VT, USA) at a wavelength of 490 nm. Results are expressed as the fraction of cell viability relative to cells cultured in physiological conditions (5 mM Glucose).

### 2.3. Detection of Apoptosis and Intracellular Levels of Nrf2

Apoptosis was evaluated through the detection of Bax and Bcl-2 proteins. Colorimetric cell-based ELISA kits (Assay Biotechnology, Sunnyvale, CA, USA) were used to measure the intracellular levels of Bax, Bcl-2, and Nrf2 proteins. Cells, seeded at 2.5 × 10^4^ cells/well in a 96-well plate, were treated as previously described (see Section 2.1). Cells were fixed in 4% formaldehyde at the end of the research. Quenching buffer, blocking buffer, and primary antibodies (rabbit polyclonal anti-Bax, rabbit polyclonal anti-Bcl-2, rabbit anti-Nrf2, and mouse monoclonal anti-GADPH) were added in this order. After an overnight incubation at 4 °C, secondary antibodies conjugated to peroxidase were added (HRP-conjugated anti-rabbit IgG for Bax, Bcl-2, and Nrf2; HRP-conjugated anti-mouse IgG for GADPH), and samples were measured at 450 nm with a microplate reader. The resulting values were standardized to GADPH OD450.

The apoptosis index (AI), defined as the ratio between Bax and Bcl-2 expression, was calculated according to the following equation:(1)$$\mathrm{AI}=\frac{\mathrm{Bax}}{\mathrm{Bcl}2}$$

### 2.4. Quantification of Intracellular Levels of ROS

ARPE-19 cells were seeded into 96-well black/clear bottom plates (Greiner Bio-One, Kremsmünster, Austria) for the determination of Reactive Oxygen Species (ROS). The 2′,7′-dichlorofluorescein diacetate (DCFDA)-Cellular ROS Detection Assay kit was used for the detection (Abcam, Cambridge, UK). DCFDA, which is initially non-fluorescent, is oxidized to DCF, a highly fluorescent molecule, and its intensity was measured using the BioTek Cytation cell imaging multimode microplate reader (BioTek U.S., Winooski, VT, USA) in end point mode at Ex/Em = 485/535 nm. The microplate reader’s imaging acquisition mode enabled the capture of fluorescence pictures of each well, which were utilized to calculate the number of cells.

ROS generation was then represented as a percentage relative to the control as the fluorescence intensity normalized to the number of cells.

### 2.5. Metabolic Imaging for the Evaluation of Intracellular Levels of NADH

Among the group of intrinsic fluorophores, Nicotinamide Adenine Dinucleotide (NAD) has been widely used as a reliable indicator of cell metabolism through the application of traditional fluorimetric experiments. However, requiring the application of cytoplasm and mitochondrial extraction and separation procedures, these techniques need chemical and mechanical manipulation that can interfere with redox processes, leading to unwanted variations of NAD(P)H redox state. To overcome these limits, since NAD in its reduced form (NADH) exhibits autofluorescence, NAD(P)H fluorescence microscopy was introduced, allowing non-invasive and real-time measurement of NAD(P)H variation [27,32,35].

Microscopy imaging was performed with a Nikon A1-MP confocal microscope, equipped with a 2-photon Ti:Sapphire laser (Mai Tai, Spectra Physics, Newport Beach, CA) producing 80-fs pulses at a repetition rate of 80 MHz. An on-stage incubator (OKOLAB S.r.l., Pozzuoli, Italy) kept the constant temperature of T = 37 °C and 5% levels of CO_2_. NADH autofluorescence, excited at 740 nm, was acquired in the spectral emission range 425–475 nm. Contributions from cytoplasm and mitochondria, respectively, have been obtained separately by applying a machine learning-based pixel classification workflow through the open-source software Ilastik (https://www.ilastik.org/, accessed on 5 May 2022) [35,36,37,38]. The separation of the cytoplasmic and mitochondrial NAD(P)H responses through microscopy [39], perfectioned through time by optimizing acquisition and segmentation techniques, is nowadays a strong and reliable assessment of NAD(P)H redox state.

### 2.6. Molecular Characterization: RNA Isolation and RT-PCR

Total RNA was isolated with the RNeasy MicroKit (Qiagen, Hilden, Germany) and RNA concentration was determined using spectrophotometric measurements at 280 and 260 nm. Total RNA was used for first strand cDNA synthesis with QuantiTect Reverse Transcription Kit (Qiagen). PowerUp™ SYBR^®^ Green (Applied Biosystem, Waltham, MA, USA) Master Mix (2X) reagents were used according to the manufacturer’s recommendations. The quantification of gene expression was obtained from 7900HT FAST REAL-TIME PCR SYSTEM (Applied Biosystems).

Primers were purchased from Thermo Fisher Scientific, Inc. (Waltham, MA, USA). Each gene target quantification reaction was performed separately with the respective primer sets, as reported in Table 1.

Despite the varied theoretical melting temperatures of each primer set, the amplification and melt curve reaction settings used a consistent annealing temperature for simultaneous operation, according to the following protocol:Stage 1 (1 cycle) at 50 °C, for 2 min, followed by 95 °C for 10 min;Stage 2 (40 cycles) at 95 °C for 15 s, followed by 60 °C for 1 min; the melt standard curve was at 95 °C for 15 s followed by 60 °C for 1 min, 95 °C for 15 s and finally 60 °C for 15 s.

Optimization and validation of primer pairs consisted of triplicate independent runs with technical triplicates, including No Template Controls (NTC) and samples for each run. Gene expression results were analyzed using the Applied Biosystem software (SDS 2.4.1, available online at https://www.thermofisher.com/it/en/home/technical-resources/software-downloads/applied-biosystems-7900ht-fast-real-timespcr-system.html, accessed on 6 May 2022). The average of the three threshold cycle values (*C_t_*) were automatically filled in with all parameters set considering the β-actin gene as an endogenous and reference control as follows:(2)$$\Delta C_{t}=C_{t}^{target}-C_{t}^{reference}$$
where $C_{t}^{target}$ is the average of the three threshold cycle values for the target gene, and $C_{t}^{reference}$ is the average threshold cycle value for the reference gene.

For relative quantification, the 2^−ΔΔCt^ method was used.

### 2.7. Statistical Analysis

Student’s t-tests for sets of biological/biophysical data were performed by Orange 3.31 (https://orangedatamining.com/, (accessed on 5 May 2022)) and R Software Version 4.1.3 (https://www.r-project.org/, (accessed on 5 May 2022)). Baseline characteristics among samples have been compared with one-way ANOVA for parametric variables. Tukey’s test was then used for post-hoc comparisons among samples.

## 3. Results

### 3.1. Effect of High Glucose on Cell Viability and Apoptosis

The effect of high-glucose concentrations and the potential therapeutic effect of DHA on the rate of cell growth of cultured ARPE-19 cells were analyzed, as described in Figure 1.

The data in Figure 1A show no significant changes in cell viability following 20 h of exposure of the cells to 50 mM glucose (HG) in comparison with the CTRL (15.0 ± 2.2% increase in viability, *p* > 0.05). Similarly, the effect of the treatment with DHA and high glucose induces no consistent variations with respect to CTRL. However, DHA alone slightly decreased cells’ viability (~−15%, *p*-value = 0.04). In view of this, we quantified the expression of both apoptosis regulator Bax and Bcl-2 (B-cell lymphoma 2) proteins by using the Bax/Bcl-2 ratio [34], which can be used as an index of apoptosis (Section 2.3). We observe a higher ratio of the protein levels Bax/Bcl-2 under high glucose conditions (0.39 ± 0.05) with respect to CTRL (0.31 ± 0.02), though this variation is non-significant (*p*-value = 0.44, Figure 1B). In contrast, DHA administration resulted in a significant decrease in the Bax/Bcl-2 balance (0.22 ± 0.01, *p*-value = 0.04), suggesting an anti-apoptotic effect. Thus, in cells treated with high glucose, DHA antagonized the high levels of Bax (keeping protein levels comparable to the untreated control) and greatly boosted the intracellular levels of Bcl-2, which HG reduces (see Appendix A). These results are consistent with the reduced values of the apoptotic index seen in control cells treated with DHA (0.23 ± 0.14) compared to CTRL (*p*-value = 0.5) and HG (*p*-value = 0.048).

### 3.2. Effect of High-Glucose and DHA on ARPE Redox Homeostasis

To evaluate the impact of high levels of glucose on the redox homeostasis and the potential ameliorative effect of DHA, we quantified the ROS production through DCFDA fluorescence intensity. Representative images of DCFDA fluorescence intensity are depicted in Figure 2A for CTRL, HG, DHA, and HG+DHA, respectively. From a qualitative comparison of the fluorescent images, it appears that in HG, green fluorescence intensity is enhanced in comparison with the CTRL, highlighting an increase in the levels of ROS. DHA that was added under high glucose conditions reduced the fluorescence intensity to that observed under the physiological conditions. The quantitative measurement of the emission intensity (Figure 2B) enables evaluating the effects of various treatments on intracellular levels of reactive oxygen species, which are given as a percentage relative to the cells maintained under physiological conditions (CTRL = 1.00). Fluorescence intensity quantification demonstrates that high-glucose causes a large rise in intracellular ROS compared to CTRL (+35%, *p*-value < 0.0001), which is greatly decreased and reduced by subsequent DHA treatment (−7% compared to HG, ° *p*-value < 0.05).

### 3.3. DHA Modulates the High Glucose-Induced Oxidative Stress by Activating the Nrf2-Nqo1-HO Signaling Pathway

The levels of nuclear factor erythroid 2-related factor gene (Nrf2) have been used to investigate the signal transduction pathways involved in the DHA-mediated therapeutic action. The relative expression with respect to CTRL, whose value has been normalized to 1, is shown in a color-coded scale ranging from 1 (brown) to 8 (dark purple) in Figure 3. From the heat-map, it is possible to observe that while Nrf2 expression levels are comparable among cells under different treatments, an increase occurs in mRNA levels of both Nqo1 and HO-1 in cells supplemented with DHA, either in physiological or HG conditions. To quantify these changes, the mean values of mRNA relative expression are reported in Figure 3B–D for Nrf2, Nqo1, and HO-1, respectively.

Figure 3B shows that neither HG nor DHA under both physiological and HG conditions stimulate the activation of Nrf2 mRNA expression, with values 1.98 ± 1.18 for HG, 1.39 ± 0.72 for DHA, and 1.48 ± 0.71 for HG+DHA respectively. To evaluate the potential transcriptional activation of Nrf2, we both assayed Nrf2 protein expression in whole cells subjected to the different treatments (see Appendix A) [40] and evaluated the expression of Nqo1 and HO-1. The normalized expression of HO-1 and NQO1 is reported in Figure 3C,D, respectively. The values show that treatment with DHA in HG conditions markedly up-regulates both the HO-1 and Nqo1 mRNA expression (with values >7-fold and >6-fold higher com- pared to CTRL), while the Nrf2/HO-1 pathway is only slightly activated in HG conditions, as well as in the presence of DHA alone. Conversely, HG has no effect on the mRNA expression of Nqo1 (1.62 ± 0.59, *p*-value = 0.95 vs. CTRL = 1.08 ± 0.50), whose activation is indeed highly enhanced by the DHA, either in physiological (5.73 ± 2.08, *p*-value = 0.003 vs. CTRL) or in high-glucose conditions (5.10 ± 1.83, *p*-value = 0.02 vs. HG alone).

### 3.4. DHA Activates the Production of Intracellular NADH to Promote the Formation of Reductive Species

Since NADH exhibits autofluorescence, we evaluated its intracellular levels and distribution in different compartments using two-photon microscopy, and results obtained for cytosolic and mitochondrial NADH, respectively, are represented in Figure 4.

Figure 4A depicts maps of cytoplasmic (on the left) and mitochondrial (on the right) NADH autofluorescence. Each pixel’s color ranges from dark purple, corresponding to low levels of NADH, to yellow, corresponding to high levels of NADH. These images show that while the intensity of mitochondrial NADH autofluorescence (second column) is comparable among CTRL, HG and HG+DHA with only a slight decrease in DHA alone, differences in the fluorescence emission of cytoplasmic NADH become evident as reported in the first column. Although no differences can be observed in HG cells with respect to CTRL, a predominance of yellow pixels emerge in HG+DHA while cells treated with DHA alone are characterized by the presence of more purple-colored pixels, representative of lower levels of NADH. A quantitative analysis of mitochondrial and cytoplasmic NADH is reported in the bar plot in Figure 4B,C, respectively. The data in Figure 4A on mitochondrial NADH indicate that the treatment with high glucose concentrations did not induce a significant change in comparison with control cells (CTRL = 2.47 ± 0.32; HG = 2.40 ± 0.22, *p*-value = 0.93). Similarly, the treatment with DHA with HG, although slightly increasing levels of mitochondrial NADH (HG+DHA = 2.71 ± 0.21) with respect to both CTRL and HG cells, did not show a significant variation compared to other cells. Interestingly, a statistically different decrease with respect to CTRL was observed in cells supplemented with DHA, with levels of mitochondrial NADH equal to 1.91 ± 0.11 (**** *p*-value < 0.0001 vs. CTRL). However, quantifying the cytosolic levels of NADH, whose values are represented in the bar plot in Figure 4B, a significant increase was retrieved in cells under high-glucose conditions treated with DHA (2.03 ± 0.11) with respect to HG alone (1.72 ± 0.11, *p*-value < 0.0001 vs. HG alone), while HG shows comparable levels with respect to CTRL (1.61 ± 0.08, *p*-value = 0.1). Again, as observed in mitochondria, DHA alone causes a significant reduction in cytoplasmic levels of NADH with respect to cells cultured in physiological conditions (1.40 ± 0.04, *p*-value = 0.001).

## 4. Discussion

Oxidative stress causes lipids, proteins, and DNA damage, as well as tissue damage, in and around retinal vessels, ultimately leading to the functional changes and breakdown of the blood–retinal barrier (BRB) [41], a particularly restrictive physiological barrier that regulates the flow of water, nutrients, ions, and metabolic waste products between the choroid and photoreceptors, whose outermost layer is constituted by the retinal pigment epithelium (RPE). DHA is a representative ω3 PUFA whose beneficial effects for health have been widely studied. In view of its well-documented protective role [42], which has been recently associated with the modulation of anti-inflammatory gene expression and activation of anti-inflammatory pathways [43,44], we focused in the present study on the investigation of the potential therapeutic effect of DHA at physiological concentration against hyperglycemia-induced oxidative damage in human RPE cells, unraveling its underlying anti-apoptotic and antioxidant effects. In our experimental model, control human RPE cells (ARPE-19) were supplemented with a 50 mM D-Glucose concentration for 36 h before measurements, and the potential therapeutic effects of DHA were investigated by adding a 60 μM physiological concentration 20 h after high-glucose treatment. Findings from our model reveal that treatment with DHA was able to counteract the high glucose-induced increase in the intracellular levels of ROS (+35%, Figure 2B) with respect to physiological culture conditions, restoring the redox homeostasis. This quenching effect is instead not observed in control cells treated with DHA, thus confirming that DHA triggers the expression of antioxidant enzymes, not being activated if ROS are not present at high levels. This result makes clear, in our experimental paradigm, that the antioxidant machinery is activated only to recover redox homeostasis itself, constituting an interesting point worthy of additional investigation to assess which are the ROS levels necessary to co-activate this pathway.

The simulated hyperglycemia did not result in a loss of cell viability (Figure 1A), with HG cells even showing an increase in cell proliferation (+15%), as already observed in other cell lines [45,46]. However, although the viability assay can provide a first indication of cell growth and death, it cannot distinguish between cellular necrosis and apoptosis, which constitute two different mechanisms that ultimately lead to cell death. While necrosis results from abrupt environmental perturbations, apoptosis is an essential and tightly controlled process, which is regulated by the interplay between opposing members of the Bcl-2 protein family: Bcl-2 and its closest homologs promote cell survival, whereas the BH3-only proteins sense and relay stress signals, activating Bax, which further permeabilizes mitochondria [47,48]. Apoptosis has been demonstrated to be activated by an excess of ROS [49]. In line with that previously observed and reported [42,50,51], our results reveal that HG induces a marked activation of the apoptotic pathway, with the increase in the apoptosis index Bax/Bcl-2 ratio, as represented in Figure 1B. Interestingly, this effect was counteracted by the treatment with DHA, 20 h after glucose administration, causing a significant shift in the balance between pro- and anti-apoptotic proteins (Figure 1B, *p*-value = 0.04), antagonizing the high levels of Bax and contextually increasing the intracellular levels of Bcl-2. Overall, these findings suggest that DHA may be able not only to protect against [42] but also to mitigate the stress-induced oxidative stress and the consequent apoptosis.

Interestingly, it has been recently observed that the nuclear factor Nrf2 and its related pathway are able to promote the activation of antioxidant cellular mechanisms [42,50]. Nrf2 constitutes a major transcription factor, which has been identified as a regulator of cellular response against oxidant stimuli, controlling the expression of a wide range of enzymes responsible for the physiological and pathophysiological outcomes of oxidant and inflammatory exposure. Among others, it encodes basic leucine zipper transcription factors, regulating the expression of antioxidant proteins and of two of its target genes, NAD(P)H quinone oxidoreductase (Nqo1) and heme-oxygenase-1 (HO-1). Our results show an almost comparable Nrf2 mRNA expression in CTRL, HG and HG+DHA cells (Figure 3B) can be explained by considering that Nrf2 is a constitutive gene whose functional activity depends on the cellular distribution between the nucleus and the cytoplasm. Indeed, under physiological conditions, that is, when a correct balance between antioxidant and prooxidant species occurs, Nrf2 is in the cytoplasm, where it establishes with Keap1 and Cullin3 an inhibitory complex for its ubiquitination and proteasome degradation [51]. Oxidative stress induces conformational modifications in Keap1 cysteine residues, which promotes the breakdown of the complex, freeing Nrf2, which is thus translocated to the nucleus, where it results in the expression of antioxidant genes, including heme oxygenase 1 (HO-1) and NAD(P)H dehydrogenase (quinone) 1 (Nqo1), which play a defensive role in cell homeostasis. Moreover, the activation of the Nrf2 pathway was also observed through the evaluation of the relative Nrf2 protein levels (Appendix A) in whole cells subjected to the different treatments under investigation. Indeed, according to recent studies [52], this “whole cell” Nrf2 protein quantification can provide an indicator of the fact that, rather than being ubiquitinated and degraded, Nrf2 is translocated from cytoplasm to nucleus, thus highlighting its activation and avoiding the requirement for further experiments to be performed on the cytoplasm and nucleus separately. This direct effect at the post-transcriptional level could also explain the rapid action in which DHA counteracts the oxidizing and apoptotic effects induced by the simulated hyperglycemia. In particular, we observed a DHA-mediated marked activation of the Nqo1 (Figure 3C) antioxidant pathway. Nqo1 constitutes one of the two major quinone reductases in mammalians, involved in cellular adaptations to oxidative stress [53,54,55,56]. Nqo1 is extremely effective at catalyzing the two-electron-mediated reduction of quinones to hydroquinones, which is commonly proposed as a mechanism of detoxification [57]. Interestingly, metabolic imaging revealed that the treatment with DHA in high-glucose conditions is able to induce a significant increase in the cytoplasmic amount of the reducing equivalent with respect to HG cells, although mitochondrial levels of NADH remain unchanged (Figure 4). This point constitutes an extremely important finding, linking Nqo1 to a modulation of the NAD+/NADH redox balance, which is differentiated between the mitochondria and cytoplasm. Though several studies in recent years have focused on the potential of Nqo1 turnover to modulate the NAD+/NADH redox balance, suggesting its potential therapeutic modulation [58], metabolic imaging provided further insights by showing the spatial distribution of these induced variations. Indeed, Nqo1 plays multiple roles in cellular adaptation to stress, and the spatial distribution of the NAD+/NADH redox balance may be a hallmark of the specific response that it triggers: apart from quinone detoxification, emerging roles of Nqo1 include its function as an efficient intracellular generator of NAD+ for enzymes, including PARP and sirtuins, and interacts with a growing list of proteins and mRNA. This suggests that Nqo1 may act as a cellular redox switch, potentially altering its interactions with other proteins and mRNA as a result of the prevailing redox environment. This change might be connected to DHAs therapeutic impact, which may initiate an adaptive protective mechanism against inflammation and oxidative stress by promoting Nqo1 expression. Other studies found that overexpression of Nqo1 in cultured vascular smooth muscle and endothelial cells protected them against pro-inflammatory cascades and oxidative cytotoxicity caused by cytokines [59,60,61]. Upregulation of Nqo1 in cultured vascular cells by adenovirus vectors also decreased the expression of hyperglycemia-mediated endothelial adhesion molecule and tumor necrosis factor -α and prevented smooth muscle cell migration [62]. A recent investigation in individuals with ischemic stroke caused by large-artery atherosclerosis discovered that Nqo1*2 polymorphisms were associated with a decreased risk of atherosclerosis-related stroke [63]. Taken together, these in-vitro and in-vivo investigations, as well as the data reported, show that gene transfer or pharmacological activation of NQO1 in the vascular or circulatory systems may constitute a promising technique for controlling vascular disorders [64]. DHA, in particular, may elicit an effective antioxidant response against hyperglycemia-induced oxidative stress and apoptotic pathways by promoting the formation of the reductive agent NADH in the cytoplasm and, therefore, maintaining the activation of the Nrf2/Nqo1 signaling cascade.

## 5. Conclusions

In conclusion, this investigation of the mechanisms underlying the impairment induced by high levels of glucose on redox homeostasis of the BRB and the subsequent recovery provided by DHA can provide both a powerful indicator for the detection of the disease as well as a potential metabolic therapeutic target for the early intervention in its treatment. The mechanisms of action and their applications in in-vivo and in-vitro systems thus deserve further investigation.

## Figures and Tables

**Figure 1 antioxidants-11-01072-f001:**
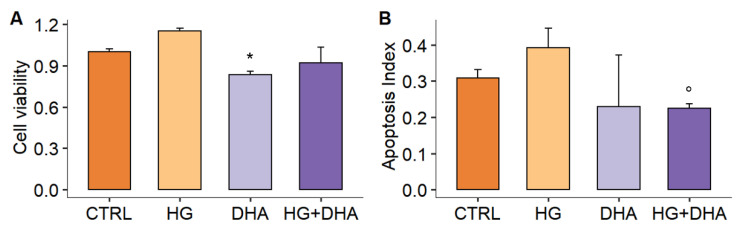
(**A**) Cell viability of ARPE–19 cultured under physiological conditions (5 mM glucose, CTRL, dark orange), with 50 mM glucose (HG, 36 h, light orange), supplemented with 60 µM DHA only (DHA, 16 h, light purple), or treated with DHA following HG administration (HG+DHA 20 h HG + 16 h DHA, dark purple). Data are expressed as a fraction of viability with respect to control cells (CTRL = 1.00 ± 0.02), and results are presented as mean ± sd (*n* = 4). * stands for *p* < 0.05 vs. CTRL). (**B**) Apoptosis index was calculated, as the Bax/Bcl-2 protein levels ratio is represented for CTRL, HG, DHA and HG+DHA cells, respectively. The value of the ratio is reported as mean ± sd on the y-axis, and statistical results obtained from Tukey’s post-hoc comparison among groups are shown along with the graph (° stands for *p* < 0.05 vs. HG alone).

**Figure 2 antioxidants-11-01072-f002:**
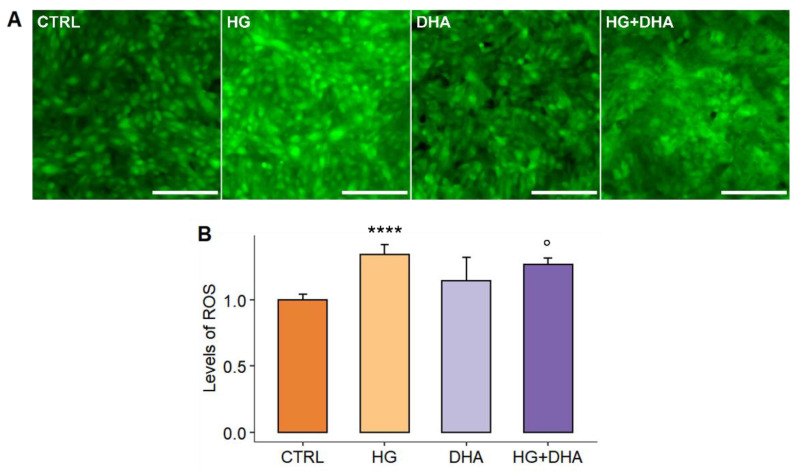
(**A**) Fluorescence intensity imaging of ROS quantified using the BioTek Cytation cell imaging multimode microplate reader. Images of green fluorescence intensity, proportional to the intracellular levels of ROS, are represented for CTRL, HG, DHA, and HG+DHA cells, respectively. Emission intensity, collected in the range of 500–550 nm, is higher in cells under HG conditions, revealing an increase in intracellular ROS, while it decreases in the presence of DHA, highlighting its modulatory effect. Images were collected with a 4× objective. Scale bar is 50 μm. (**B**) Intracellular levels of ROS are represented for CTRL, HG, DHA, and HG+DHA cells, respectively. The value of intracellular ROS is reported as mean ± sd (*n* = 100 cells) on the y-axis. Statistical results obtained from Tukey’s post-hoc comparison among groups are shown along with the bar plot (**** stands for *p* < 0.0001 vs. CTRL; ° stands for *p* < 0.05 vs. HG alone).

**Figure 3 antioxidants-11-01072-f003:**
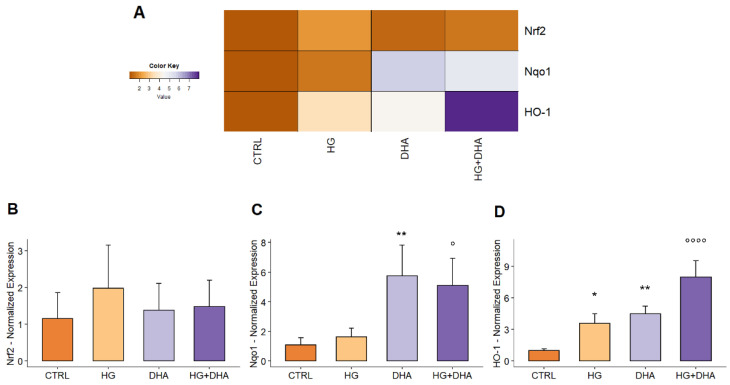
(**A**) Heatmap showing the expression of different genes normalized with respect to CTRL = 1. Data are displayed in a grid, where each row represents a gene, and each column represents a sample. Boxes are colored according to the color key bar represented along with the heat-map, ranging from 1 (CTRL gene expression, in brown) to 8 (high gene up-regulation, in purple). The color of the boxes provides an indication of changes in mRNA expression. The quantitative normalized mRNA expression is represented for CTRL, HG, DHA, and HG+DHA cells, in (**B**) for Nrf2, in (**C**) for Nqo1, and in (**D**) for HO-1, respectively. Values of mRNA expression with respect to control cells (CTRL = 1) are reported as mean ± sd on the y-axis. Statistical results obtained from Tukey’s post-hoc comparison among groups are shown along with the bar plot (* stands for *p* < 0.05 vs. CTRL; ** stands for *p* < 0.01 vs. CTRL; ° stands for *p* < 0.05 vs. HG alone; °°°° stands for *p* < 0.0001 vs. HG alone).

**Figure 4 antioxidants-11-01072-f004:**
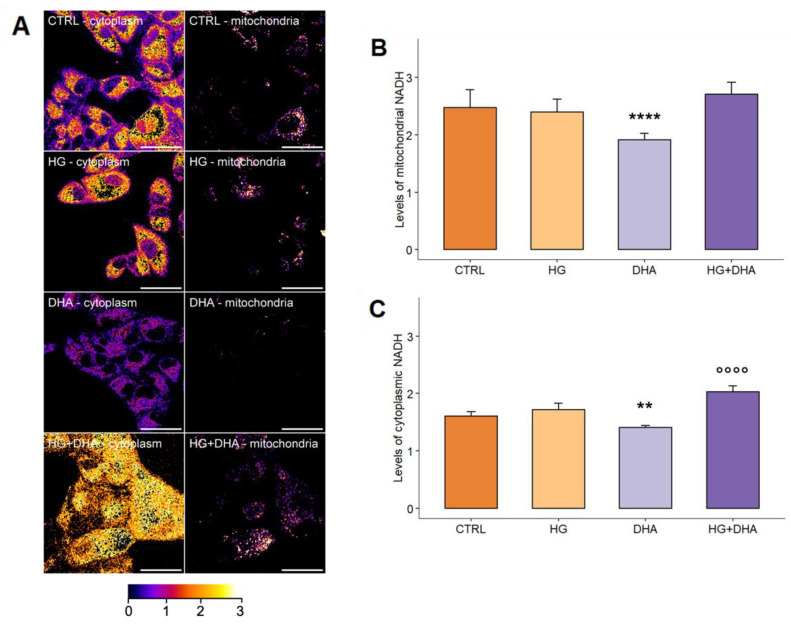
(**A**) Representative confocal microscopy images of cytoplasmic (on the left) and mitochondrial (on the right) NADH autofluorescence for CTRL, HG, DHA, and HG+DHA ARPE-19 cells. Pixel colors range from dark purple (low levels of NADH) to yellow (high levels of NADH), according to the color bar reported along with images. Scale bar is 50 μm. In the bar plot, the quantification of intracellular levels of mitochondrial (**B**) and cytoplasmic (**C**) NADH is reported as mean ± sd (*n* = 40 cells) on the y-axis for CTRL (dark orange), HG (light orange), DHA (light purple), and HG+DHA (dark purple) cells, respectively. Statistical results obtained from Tukey’s post-hoc comparison among groups are shown along with the bar plot (** stands for *p*-value < 0.01 vs. CTRL; **** stands for *p* < 0.0001 vs. CTRL; °°°° stands for *p* < 0.0001 vs. HG alone).

**Table 1 antioxidants-11-01072-t001:** List of primers used for quantitative RT-PCR.

Gene Target	Accession Code	Primer Sequence Forward (5′ to 3′)	Primer Sequence Reverse (5′ to 3′)
Β-Actin	NM_001101.5	AAACTGGAACGGTGAAGGTG	GTGGCTTTTAGGATGGCAAG
Nrf2	NM_006164.4	GTCACATCGAGAGCCCAGTC	ACCATGGTAGTCTCAACCAGC
Nqo1	X06985.1	GGTTTGAGCGAGTGTTCATAGG	CAGAGAGTACATGGAGCCAC
HO-1	J03934.1	CTGGAGGAGGAGATTGAGCG	ATGGCTGGTGTGTAGGGGAT

## Data Availability

The data are contained within the article and Appendix A.

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
