# Peer review of "Investigation of DHA-Induced Regulation of Redox Homeostasis in Retinal Pigment Epithelium Cells through the Combination of Metabolic Imaging and Molecular Biology"

_antioxidants, 2022, doi:10.3390/antiox11061072_

Round 1
Reviewer 1 Report
1. Authors claim that high-glucose conditions activated erythroid 2-related factor Nrf2, and this is the pathogenic cause. In-fact, its not the activation, but its localization in oxidative conditions (Cytoplasm to Nucleus translocation) which is the main cause of increasing Nqo1.
It would be needed to show this translocation of Nrf2.
2. Fig 2A: if ROS activates Nrdf2 and Nqo1 and Ho1 expression, one would expect that DHA would quench ROS ?? this is not observed
3. Fig: 3a Antioxidant regulatory elements (ARE) genes can be regulated by other factors too, cannot assume its Nrf2.
4. It is recommended that authors perform a traditional spectroscopy to measure cytoplasmic and mitochondrial NADH, rather than relying on machine learning with out staining controls.
5. Fig 2 A is missing scale bars.
Reviewer 2 Report
This study explored the possible utility of a DHA product with antioxidant activity on retinal pigment epithelium cells. This paper is fascinating and well-organized. I recommend it for publication in the Antioxidants.
I think that extensive editing of the English language and style is required.
for example:
- line 66 - change for "factors leading to the etiology of diabetic retinopathy"
- line 71 - change for "water, nutrients, ions, and waste products flux into and out of the retinal compartment"
- line 75 - change for "essential for cells' survival and proper function in the visual process"
etc.
Round 2
Reviewer 1 Report
Authors have provided answers to all my comments. I recommend acceptance and congrats on a very nice study.